# A multi-level analysis of prevalence and factors associated with caesarean section in Nigeria

Kobi V. Ajayi[1,2], Samson Olowolaju[3], Yusuf H. Wada[4], Sonya Panjwani[2], Bright Ahinkorah[5], Abdul-Aziz Seidu[6,7], Collins Adu[7,8,9], Olajumoke Tunji-Adepoju[10], Obasanjo Afolabi Bolarinwa[11,12,13]*

1 Educating Directing Empowerment & Nurturing (EDEN) Foundation, Abuja, Nigeria, 2 Department of Health Behaviour, School of Public Health, Texas A&M University, College Station, Texas, United States of America, 3 Department of Demography, College for Health, Community and Policy, the University of Texas at San Antonio, San Antonio, Texas, United States of America, 4 Society for Family Health, Abuja, Nigeria, 5 School of Public Health, Faculty of Health, University of Technology Sydney, Sydney, Australia, 6 Centre for Gender and Advocacy, Takoradi Technical University, Takoradi, Ghana, 7 Public Health & Tropical Medicine, College of Public Health, Medical and Veterinary Sciences, James Cook University, Townsville, Australia, 8 Department of Health Promotion and Disability Studies, Kwame Nkrumah University of Science and Technology, Kumasi, Ghana, 9 Centre for Social Research in Health, University of New South Wales, Sydney, New South Wales, Australia, 10 Department of Sociology, University of Ibadan, Ibadan, Nigeria, 11 Department of Public Health & Well-being, Faculty of Health, and Social Care, University of Chester, Chester, United Kingdom, 12 Institute for Advanced Studies in the Humanities, University of Edinburgh, Edinburgh, United Kingdom, 13 Discipline of Public Health Medicine, School of Nursing and Public Health, University of KwaZulu-Natal, Durban, South Africa

* bolarinwaobasanjo@gmail.com

**Data Availability Statement:** The datasets utilized in this study can be accessed at https://dhsprogram.com/data/available-datasets.cfm.

## Abstract

The choice of caesarean section (CS) plays a significant role in maternal and neonatal health. However, suboptimal CS uptake suggests unmet obstetric care leading to adverse maternal and neonatal health. Considering that maternal health problems in Nigeria remain a public health problem, this present study aims to assess the prevalence and multilevel factors associated with caesarean section among women of reproductive age in Nigeria. Data from the 2018 Nigeria Demographic and Health Survey were analysed. Our analyses included 19,964 women of reproductive age, with their last birth within five years preceding the survey. Multilevel logistic regression analysis was carried out to examine the predictors of the caesarean section in Nigeria. The prevalence of CS among women of reproductive age in Nigeria was 3.11%. Women from the Yoruba ethnic group [aOR = 0.52; 95%(CI = (0.32–0.84)], with two children [aOR = 0.67; 95%(CI = 0.52–0.88)], three children [aOR = 0.49; 95%(CI = 0.36–0.66)], four children and above [aOR = 0.34; 95%(CI = 0.26–0.46)], those who practised Islam [aOR = 0.74; 95%(CI = (0.56–0.99)], and those that had a normal weighted baby [aOR = 0.73; 95%(CI = 0.60–0.99)] were less likely to report having a CS in Nigeria compared to those from Hausa/Fulani ethnic group, those who had one child, those who practised Christianity, and those who had a high weighted baby. Also, women residing in rural areas [aOR = 0.79; 95% (CI = (0.63–0.99)] and the South-South [aOR = 0.65; 95% (CI = (0.46–0.92)] were less likely to have CS compared to those residing in urban areas and North Central. The study concluded that several individual and community-level factors,

**Funding:** The authors received no specific funding for this work.

**Competing interests:** The authors have declared that no competing interests exist.

such as religious belief, number of children, ethnicity, place of residence, and region of residence, were associated with CS utilisation in Nigeria. Our study highlights the need for different regional, local, and cultural contexts for evidence-based policy and programmatic efforts to facilitate equitable access to a caesarean section in Nigeria.

## Background

Globally, there has been a steady upsurge in the use of caesarean section (CS). Between 1990 to 2014, the global average use of CS increased from 12.4% to 18.6% [1], exceeding the World Health Organization's (WHO) recommended acceptable range of 10–15% [2]. Although there is no evidence that CS benefits maternal and infant health beyond the WHO upper limit, use below 10% is inadequate and suggests unmet emergency obstetric care [2]. Yet, CS use differs, with higher use of up to 27.2% found in developed nations than in the least developed nations (6%) [1]. More specifically, the prevalence of CS in Africa and sub-Saharan Africa (SSA) is 7.3% and 4.9%, respectively [1, 3]. Although there is a wide variation of CS use in SSA, Nigeria has a considerably lower CS use prevalence of 2.1% than her neighbouring countries [3–5]. Given that the CS rate in Nigeria is well below the WHO recommendations, it is evident that there is a substantial unmet obstetric need in the country, which scholars assert is associated with adverse maternal and infant health outcomes in the country [6].

Extensive evidence reports significant adverse maternal and neonatal health indexes in Nigeria. For example, the country accounts for almost 20% of global maternal mortality [7]. The country also has an infant mortality rate of 72.2 per 1000 live births and an under-five mortality rate of 113.8 per 1000 live births [8, 9], all of which are higher than most other African countries [10]. As such, scaling access to CS in Nigeria is central to achieving Sustainable Development Goals (SDGs). Specifically, Goal 3, Target 3.1, seeks to reduce the global maternal mortality ratio to less than 70 per 100,000 live births, while Target 3.2 aims to end preventable neonatal and under-five mortality to as low as 12 per 1000 live births and 25 per 1000 live births, respectively [11].

The advantages of CS have been well documented [2]. In some cases, CS is the most feasible option in high-risk pregnancies, such as multiple pregnancies, breech birth, obstructed labours, chronic conditions such as high blood pressure, and the prevention of transmissible infections such as HIV/AIDS [2]. This supports the premise that greater access to CS indicates optimal obstetric care access, resulting in improved maternal and neonatal health. As a result, improving access to CS utilisation is critical to achieving favourable maternal and neonatal health in Nigeria [6].

Research on CS utilisation in Nigeria revealed a significant association between CS and age, place of residence, socioeconomic status, educational level, birth order, birth weight, obstetric complications, religious beliefs, and having a previous CS [4, 12–15]. However, most studies rely on institutional-based data, which are limited to sample size bias [12–14]. Nonetheless, to the best of our knowledge, despite the existing nationally representative studies [4, 6, 16], none have theoretically provided a comprehensive investigation of the multilevel factors predicting CS utilisation in Nigeria. According to the Andersen behavioural model of healthcare utilisation, healthcare access and uptake are influenced by several individual (e.g., age, marital status) and contextual factors (e.g., residential area, income levels) embedded within the predisposing, enabling, and need levels [17]. Together, these factors can facilitate or impede healthcare utilisation, such as CS uptake. Understanding the multilevel factors is advantageous because it

considers the variability within nested data and has powerful estimation. Thus, this current study is important because it will enable policymakers and other relevant stakeholders to identify multilevel factors associated with CS utilisation. It will also help to create strategies to effectively address them from a national policy standpoint. For example, programs that promote maternal health care utilisation by addressing proximal (e.g., health care seeking attitudes during pregnancy) and distal factors (e.g., education of skilled birth attendants) will go a long way in addressing multilevel determinants influencing knowledge and uptake of CS, thereby improving maternal and infant health in Nigeria [18]. Therefore, this study builds on previous studies by investigating the multilevel factors associated with CS utilisation and the prevalence of CS in Nigeria. The study aims to answer the research question: What are the prevalence and the multilevel factors associated with CS in Nigeria?

## Methods and materials

### Data source

This study analysed the latest Nigeria Demographic and Health Survey (NDHS) dataset conducted in the year 2018 [19]. NDHS is a nationally representative survey that provides up-to-date information on demographic and health indicators. NDHS utilises a stratified, two-stage cluster design in selecting its samples. The sampling units for the first stage were the country's enumeration areas (EAs), while the second stage included a complete listing of households carried out in each of the 1,400 selected EAs. The NDHS targets women of childbearing age (15–45) and men aged 15–59 in randomly selected households across Nigeria. Accordingly, a sample of approximately 42,000 households was selected for the survey, presumed to represent the Nigerian population concerning the selected demographic and health indicators.

The NDHS file used has socioeconomic and demographic characteristics, including maternal age at last birth and mode of delivery among women of reproductive age whose previous birth was within five years preceding the survey. Our analyses only included a total of 19,964 women of reproductive age whose last delivery was within five years preceding the survey. Only reproductive aged women with at least one child and/or who had any delivery in the last five years were included in this study. The exclusion criteria were women who had no child or had not had any delivery in the last five years and/or were pregnant for the first time at the time of the survey. Detailed information about the sampling frame, method, and survey design are reported elsewhere [19].

### Description and definition of variables

**Outcome variable.** Similar to previous studies [4, 16], the outcome variable was the mode of delivery among women of reproductive age, coded as binary. Specifically, women responded to questions about whether they gave birth through CS or not. We then coded 0 "No" (indicating that the last birth was non-CS) and 1 "Yes" (indicating that the last birth was via CS). Our category of interest was women who responded in the affirmative to having a CS.

**Individual-level explanatory variables.** The individual level explanatory variables used in this study were considered empirically [4, 6, 20, 21]. They include maternal age, maternal educational level, partner educational level, marital status, occupation, ethnicity, parity, religion, mass media exposure, child's sex, child's weight at birth, and the number of antenatal care visits. Maternal age was recoded into three age groups (15–24, 25–34, and 35+). Maternal and partner educational levels were recoded into three groups (No education, Primary, and secondary & higher). Marital status was recoded into two categories (currently married; cohabiting). Employment status was grouped as currently employed and not currently employed. We regrouped the ethnicity variable into four categories (Hausa/ Fulani, Yoruba, Igbo, and

others). The Fulani ethnic group was combined with the Hausa ethnic group, while other groups that were not these three major ethnic categories were classified as others.

Parity was recoded into four categories (1, 2, 3, and 4+). We categorised religion into three groups (Christianity, Islam, and traditional & others). To obtain the mass media exposure variable, we first regrouped the three NDHS variables that asked questions on the frequency of reading newspapers, listening to the radio, and watching television into two groups—the groups that performed the media activities at least once a week and those that did not. Hence, a score of 1 was assigned to individual women that engaged in any of the three media activities at least once a week. In comparison, a score of zero was assigned to women that did not engage in any of the three media activities at least once a week. Women with a score of 1 were categorised as having media exposure, and women with a score of 0 were classified as not having media exposure. The child's sex was categorised into male and female. The childbirth weight variable was recoded into three categories (low, normal, or high). In line with previous studies, this categorisation is a proxy for birth weight [21, 22]. Finally, we recoded the ANC visits into three groups (0, 1–3, and 4+).

**Household/Community level explanatory variables.** Similar to the individual-level explanatory variables, the household/community level factors examined in this study were empirically identified [23, 24]. The household level variables were place of residence (rural vs urban region), wealth index, and household head's sex. The community level variables were community literacy level and community-level socioeconomic status. Although these variables were listed in the NDHS, the community literacy level and socioeconomic status are not provided. Thus, we aggregated the individual mother's characteristics within the cluster [24]. The mean values of the aggregated clusters were then divided into groups based on the national mean value. Hence, community literacy level was derived from the individual-level educational status within the clusters or community. Also, the community level of socioeconomic status was derived from the mean values of individual-level wealth index categories in the community. These variables were grouped as low, medium, and high [24].

**Data analyses.** The weighted summary statistics of socioeconomic and demographic variables were computed and stratified by the dependent variable (CS). Chi-square test was used to assess the independence of the outcome variable on the different classifications of the explanatory variables. All variables with a chi-square p-value less than 0.05 were added to the multilevel analysis. We examined the association between the explanatory variables (Individual and household/community-level) and CS using multilevel logistic regression.

A multilevel regression analysis, with mixed and random effects, was applied to this current study because of its features of dealing with a hierarchical structure with multiple levels of influence on the outcome variable [25, 26]. Five models, including the null model (Model 0), were fitted. The null model examined the variation in CS across the communities without the influence of the explanatory variables. Model 1 estimates the effect of the individual-level explanatory variables on CS. Model 2 included only the household-level variables in estimating the influence of CS.

On the other hand, Model 3 estimates the effect of community-level variables on CS. The last model, Model 4, included both individual and household/community-level variables in estimating the influence of CS. The fixed effect of the models was estimated using adjusted Odds Ratios (aOR) with a 95% confidence interval. The random effect variations of the models, on another hand, were derived by using the Intra-Cluster Correlation Coefficient [27, 28] to reveal the importance of cluster surveys that used household/community level characteristics [29, 30].

The ICC is determined by the ratio of the variance of interest to the total variance, which can be expressed as ICC = (variance of interest) / (variance of interest + unwanted variance).

When the unwanted variance, such as the variance between subjects, is greater than or equal to the variance of interest, the method's reliability is considered to be poor and considered to be good if otherwise [28].

We tested for multicollinearity using variance inflation factor (VIF), which showed no evidence of collinearity among the independent variables with mean VIF = 1.85, Maximum VIF = 2.78, and Minimum VIF = 1.05. The significance level was set at 5%. All data analyses were weighted and conducted using Stata 14.1 for Mac OS (College Station, TX).

### Ethics approval and consent to participate

Since the authors of this manuscript did not collect the data, we sought permission from the MEASURE DHS website and access to the data was provided after our intent for the request was assessed and approved. More details about data and ethical standards are available at: http://goo.gl/ny8T6X.

## Results

### Sociodemographic characteristics of respondents

A total of 19,964 women of reproductive age between 15 to 49 years were included in the study. At the individual level, 9,643 (48.30%) of the respondents were aged 25 to 34. About 9,351 (46.84%) had no education, while 9,673 (48.45%) of the respondents' partners had secondary education and above. About 13,544 (67.84%) of the respondents were employed, and 12,281 (61.52%) had mass media exposure. About 10,251 (51.35%) of childbirth weight was normal, and for women with four or more ANC visits was 11,460 (57.40%) (Table 1).

At the household/community- level, 12,297 (61.59%) of the study respondents resided in rural areas, and males headed 18,590 (93.11%) households. 6,818 (34.15%) were from a community with high literacy level, while 12,097 (60.59%) were from low socioeconomic status. All the individual and household/community factors were significantly associated with CS in Nigeria except marital status and child's sex. The prevalence of CS among women of reproductive age in Nigeria was 3.11% (Table 1).

### Multilevel fixed effects (measures of associations)

The factors associated with CS at the individual level include maternal age, ethnicity, parity, religion, childbirth weight, and antenatal care visits (Table 2).

The likelihood of having CS was high among women age 25–34 [aOR = 1.67; 95% (CI = 1.23–2.27)], 35 & above [aOR = 3.56; 95%(CI = 2.50–5.06)], those that had 1–3 visits [aOR = 1.91; 95%(CI = 1.01–3.62)], and 4 & above visits [aOR = 3.98; 95%(CI = 2.27–6.98)] compared with women who were age 15 to 24 and women without ANC visit.

On the other hand, women of reproductive age (25–34 years and 35 and above) who were from the Yoruba ethnic group [aOR = 0.52; 95%(CI = (0.32–0.84)], those who had two children [aOR = 0.67; 95%(CI = 0.52–0.88)], three children [aOR = 0.49; 95%(CI = 0.36–0.66)], four and above children [aOR = 0.34; 95%(CI = 0.26–0.46)], those who practice Islam [aOR = 0.74; 95%(CI = (0.56–0.99)], and those that had normal baby weight [aOR = 0.73; 95% (CI = 0.60–0.99)] were less likely to report having CS compared to those from Hausa/ Fulani ethnic, those who had one child, those who practice Christianity, and those who had high baby weight.

At the household level, the factors associated with CS in Nigeria were the place of residence, wealth index, and region. Women currently within the middle wealth index [aOR = 2.60; 95% (CI = (1.38–4.92)], richer wealth index [aOR = 3.34; 95%(CI = (1.72–6.47)], and richest wealth

**Table 1. Distribution of individual & household/community factors among women and prevalence of caesarean section in Nigeria, NDHS 2018.**

| Variable (19,964) | Weighted Frequency | Weighted Percentage | Caesarean Section | | p-value ($\chi^2$) |
|---|---|---|---|---|---|
| | | | No | Yes | |
| **Individual level** | | | | | |
| **Maternal age** | | | | | p<0.001 |
| 15–24 | 4,808 | 24.08 | 98.66 | 1.34 | |
| 25–34 | 9,643 | 48.30 | 96.82 | 3.18 | |
| 35 & above | 5,513 | 27.61 | 95.48 | 4.52 | |
| **Maternal educational level** | | | | | p<0.001 |
| No Education | 9,351 | 46.84 | 99.33 | 0.67 | |
| Primary Education | 2,906 | 14.56 | 98.37 | 1.63 | |
| Secondary & above | 7,707 | 38.60 | 93.38 | 6.62 | |
| **Partner educational level** | | | | | p<0.001 |
| No Education | 7,569 | 36.91 | 99.48 | 0.52 | |
| Primary | 2,723 | 13.64 | 97.48 | 2.52 | |
| Secondary & above | 9,672 | 48.45 | 94.70 | 5.30 | |
| **Marital Status** | | | | | 0.93 |
| Currently married | 19,363 | 96.99 | 96.89 | 3.11 | |
| Cohabiting | 601 | 3.01 | 96.98 | 3.02 | |
| **Currently working** | | | | | p<0.001 |
| No | 6,421 | 32.16 | 97.86 | 2.14 | |
| Yes | 13,543 | 67.84 | 96.44 | 3.56 | |
| **Ethnicity** | | | | | p<0.001 |
| Hausa/Fulani | 9,283 | 46.50 | 99.05 | 0.95 | |
| Yoruba | 2,371 | 11.88 | 94.40 | 5.59 | |
| Igbo | 2,335 | 11.70 | 90.89 | 9.11 | |
| Others | 5,975 | 29.93 | 96.87 | 3.13 | |
| **Parity** | | | | | p<0.001 |
| 1 | 3,107 | 15.56 | 95.03 | 4.97 | |
| 2 | 3,597 | 18.02 | 95.73 | 4.27 | |
| 3 | 3,060 | 15.33 | 95.78 | 4.22 | |
| 4 & above | 10,200 | 51.09 | 98.20 | 1.80 | |
| **Religion** | | | | | p<0.001 |
| Christianity | 7,084 | 35.48 | 93.61 | 6.39 | |
| Islam | 12,783 | 64.03 | 98.70 | 1.30 | |
| Traditionalist & others | 97 | 0.49 | 98.79 | 1.21 | |
| **Mass media exposure** | | | | | p<0.001 |
| Not Exposed | 7,683 | 38.48 | 99.37 | 0.63 | |
| Exposed | 12,281 | 61.52 | 95.34 | 4.66 | |
| **Child's sex** | | | | | 0.60 |
| Male | 10,210 | 51.14 | 96.81 | 3.19 | |
| Female | 9,754 | 48.86 | 96.98 | 3.02 | |
| **Childbirth weight** | | | | | p<0.05 |
| High | 6,965 | 34.89 | 96.38 | 3.62 | |
| Normal | 10,252 | 51.35 | 97.16 | 2.84 | |
| Low | 2,747 | 13.76 | 97.21 | 2.79 | |
| **Antenatal care visits** | | | | | p<0.001 |
| No visit | 4,962 | 24.85 | 99.74 | 0.26 | |
| 1–3 visits | 3,542 | 17.74 | 98.89 | 1.11 | |

*(Continued)*

**Table 1.** (Continued)

| Variable (19,964) | Weighted Frequency | Weighted Percentage | Caesarean Section | | p-value ($\chi^2$) |
|---|---|---|---|---|---|
| | | | No | Yes | |
| 4 & above visits | 11,460 | 57.40 | 95.04 | 4.96 | |
| **Household-level** | | | | | |
| **Place of residence** | | | | | p<0.001 |
| Urban | 7,668 | 38.41 | 94.13 | 5.87 | |
| Rural | 12,296 | 61.59 | 98.61 | 1.39 | |
| **Wealth index** | | | | | p<0.001 |
| Poorest | 4,468 | 22.39 | 99.68 | 0.32 | |
| Poorer | 4,494 | 22.51 | 99.27 | 0.72 | |
| Middle | 4,026 | 20.16 | 97.98 | 2.02 | |
| Richer | 3,628 | 18.17 | 96.31 | 3.69 | |
| Richest | 3,348 | 16.77 | 89.30 | 10.70 | |
| **Region** | | | | | p<0.001 |
| North Central | 2,786 | 13.96 | 97.04 | 2.96 | |
| North East | 3,613 | 18.10 | 98.78 | 1.22 | |
| North West | 7,409 | 37.11 | 99.11 | 0.889 | |
| South East | 1,794 | 8.99 | 93.61 | 6.39 | |
| South South | 1,645 | 8.24 | 93.74 | 6.26 | |
| South West | 2,717 | 13.61 | 92.27 | 7.73 | |
| **Sex of household head** | | | | | p<0.05 |
| Male | 18,590 | 93.11 | 97.00 | 3.00 | |
| Female | 1,374 | 6.89 | 95.38 | 4.62 | |
| **Community level** | | | | | |
| **Community literacy level** | | | | | p<0.001 |
| Low | 6,663 | 33.37 | 99.62 | 0.38 | |
| Medium | 6,484 | 32.48 | 98.38 | 1.62 | |
| High | 6,817 | 34.15 | 92.82 | 7.18 | |
| **Community socioeconomic status** | | | | | p<0.001 |
| Low | 12,097 | 60.59 | 99.02 | 0.98 | |
| Medium | 823 | 4.12 | 96.93 | 3.07 | |
| High | 7,044 | 35.29 | 93.23 | 6.77 | |
| **Total** | **19,964** | **100** | **96.89** | **3.11** | |

Source: NDHS, 2018

index [aOR = 6.18; 95%(CI = (3.09–12.35)] were more likely to have CS. On the other hand, women residing in rural areas [aOR = 0.79; 95% (CI = (0.63–0.99)] and South-South [aOR = 0.65; 95% (CI = (0.46–0.92)] were less likely to have CS compared to those residing in urban areas and North Central. The community level results showed that women who were currently residing in areas with high community literacy levels [aOR = 1.90; 95% (CI = (1.12–3.21)] were more likely to report having CS compared to the poorest women and those residing in a community with low literacy level (Table 2).

## Random effects (measures of variations)

The empty model (Model 0) in Table 2 depicts a substantial variation in the likelihood of having CS across the Primary Sampling Units clustering [σ2 = 2.20; 95%(CI = 1.72–2.81)]. Model

**Table 2. Multilevel logistic regression models for individual, household, and community factors associated with caesarean section among women in Nigeria, NDHS 2018.**

| Variables (19,964) | Model 0 | Model 1 | Model 2 | Model 3 | Model 4 |
|---|---|---|---|---|---|
| **Individual level** | | aOR [95% CI] | aOR [95% CI] | aOR [95% CI] | aOR [95% CI] |
| **Maternal age** | | | | | |
| 15–24 | | 1 | | | 1 |
| 25–34 | | 2.17***[1.61–2.94] | | | 1.67**[1.23–2.27] |
| 35 & above | | 5.15***[3.64–7.29] | | | 3.56***[2.50–5.06] |
| **Maternal educational level** | | | | | |
| No Education | | 1 | | | 1 |
| Primary Education | | 1.16[0.75–1.79] | | | 0.89[0.58–1.38] |
| Secondary & above | | 2.31***[1.57–3.39] | | | 1.29[0.87–1.91] |
| **Partner educational level** | | | | | |
| No Education | | 1 | | | 1 |
| Primary | | 1.49[0.94–2.37] | | | 1.38[0.87–2.20] |
| Secondary & above | | 1.84**[1.21–2.81] | | | 1.27[0.83–1.95] |
| **Marital Status** | | | | | |
| Currently married | | 1 | | | 1 |
| Cohabiting | | 0.54*[0.32–0.92] | | | 0.62[0.36–1.06] |
| **Currently working** | | | | | |
| No | | 1 | | | 1 |
| Yes | | 1.06[0.85–1.34] | | | 1.13[0.90–1.42] |
| **Ethnicity** | | | | | |
| Hausa/Fulani | | 1 | | | 1 |
| Yoruba | | 0.98[0.66–1.44] | | | 0.52**[0.32–0.84] |
| Igbo | | 1.51*[1.00–2.28] | | | 1.38[0.84–2.26] |
| Others | | 1.04[0.73–1.49] | | | 0.83[0.56–1.23] |
| **Parity** | | | | | |
| 1 | | 1 | | | 1 |
| 2 | | 0.65**[0.50–0.85] | | | 0.67**[0.52–0.88] |
| 3 | | 0.46***[0.34–0.61] | | | 0.49***[0.36–0.66] |
| 4 & above | | 0.29***[0.22–0.38] | | | 0.34***[0.26–0.46] |
| **Religion** | | | | | |
| Christianity | | 1 | | | 1 |
| Islam | | 0.82[0.62–1.09] | | | 0.74*[0.56–0.99] |
| Traditionalist & others | | 0.63[0.14–2.75] | | | 0.59[0.13–2.57] |
| **Mass media exposure** | | | | | |
| Not Exposed | | 1 | | | 1 |
| Exposed | | 1.74***[1.28–2.35] | | | 1.20[0.88–1.65] |
| **Child's sex** | | | | | |
| Male | | 1 | | | 1 |
| Female | | 0.90[1.28–2.35] | | | 0.89[0.75–1.06] |
| **Childbirth weight** | | | | | |
| High | | 1 | | | 1 |
| Normal | | 0.71***[0.58–0.86] | | | 0.73**[0.60–0.88] |
| Low | | 1.10[0.82–1.46] | | | 1.18[0.89–1.56] |
| **ANC visits** | | | | | |
| No visit | | 1 | | | 1 |
| 1–3 visits | | 2.21*[1.17–4.18] | | | 1.91*[1.01–3.62] |

(*Continued*)

**Table 2.** (Continued)

| Variables (19,964) | Model 0 | Model 1 | Model 2 | Model 3 | Model 4 |
|---|---|---|---|---|---|
| 4 & above visits | | 5.24***[3.00–9.17] | | | 3.98***[2.27–6.98] |
| **Household-level** | | | | | |
| **Place of residence** | | | | | |
| Urban | | | 1 | | 1 |
| Rural | | | 0.68**[0.54–0.85] | | 0.79*[0.63–0.99] |
| **Wealth index** | | | | | |
| Poorest | | | 1 | | 1 |
| Poorer | | | 2.58**[1.35–4.92] | | 1.70[0.88–3.27] |
| Middle | | | 5.32***[2.90–9.77] | | 2.60**[1.38–4.92] |
| Richer | | | 8.54***[4.66–15.67] | | 3.34***[1.72–6.47] |
| Richest | | | 22.03***[12.01–40.41] | | 6.18***[3.09–12.35] |
| **Region** | | | | | |
| North Central | | | 1 | | 1 |
| North East | | | 0.63*[0.43–0.93] | | 0.74[0.51–1.10] |
| North West | | | 0.35***[0.22–0.51] | | 0.45***[0.29–0.71] |
| South East | | | 1.15[0.84–1.57] | | 0.43***[0.29–0.66] |
| South South | | | 0.90[0.65–1.27] | | 0.65*[0.46–0.92] |
| South West | | | 0.95[0.70–1.29] | | 0.91[0.64–1.31] |
| **Sex of household head** | | | | | |
| Male | | | 1 | | 1 |
| Female | | | 1.16[0.87–1.55] | | 1.03[0.77–1.38] |
| **Community level** | | | | | |
| **Community literacy level** | | | | | |
| Low | | | | 1 | 1 |
| Medium | | | | 3.27***[2.04–5.24] | 1.33[0.81–2.17] |
| High | | | | 10.03***[6.22–16.18] | 1.90*[1.12–3.21] |
| **Community socioeconomic status** | | | | | |
| Low | | | | 1 | 1 |
| Medium | | | | 1.95*[1.13–3.35] | 1.32[0.80–2.19] |
| High | | | | 2.39***[1.80–3.16] | 0.90[0.66–1.25] |
| **Random effects results** | | | | | |
| PSU Variance (95% CI) | 2.20[1.72–2.81] | 0.47[0.29–0.77] | 0.46[0.28–0.77] | 0.67[0.45–0.99] | 0.24[0.10–0.54] |
| ICC | 0.40 | 0.13 | 0.12 | 0.17 | 0.07 |
| LR Test | $\chi^2$ = 256.90, p<0.001 | $\chi^2$ = 26.81, p<0.01 | $\chi^2$ = 24.64, p<0.001 | X2 = 49.00, p<0.01 | $\chi^2$ = 7.42. p<0.01 |
| Wald $\chi^2$ | Reference | 495.78*** | 451.62*** | 322.99*** | 619.99*** |
| **Model fitness** | | | | | |
| Log-likelihood | -2,471.58 | -2163.62 | -2214.99 | -2283.49 | -2088.68 |
| AIC | 4,947.15 | 4375.24 | 4455.99 | 4578.98 | 4255.35 |
| BIC | 4,962.94 | 4564.63 | 4558.57 | 4626.32 | 4563.10 |

(*Continued*)

**Table 2.** (Continued)

| Variables (19,964) | Model 0 | Model 1 | Model 2 | Model 3 | Model 4 |
|---|---|---|---|---|---|
| Number of clusters | 1,387 | 1,387 | 1,387 | 1387 | 1,387 |

Weighted NDHS, 2018

PSU = Primary Sampling Unit; LR Test = Likelihood ratio Test; AIC = Akaike's Information Criterion; BIC = Schwarz's Bayesian Information Criteria; ICC = Intra-Class Correlation.

Exponentiated coefficients; 95% confidence intervals in brackets; aOR = Adjusted Odds Ratios; CI = Confidence Interval; 1 = Reference Category

*p< 0.05

**p< 0.01

***p< 0.001.

Model 0 is the null model, a baseline model without any explanatory variable included.

Model I is adjusted for individual-level variables (maternal age, maternal educational level, partner's educational level, marital status, currently working ethnicity, parity, religion, mass media exposure, child's sex, child birth weight, and ANC visits).

Model II is adjusted for household/community level variables (place of residence, wealth index, region, sex of household head, community literacy level and community socioeconomic status).

Model III is the final model adjusted for both individual and household/community variables.

0 indicated that 40% of the variation in CS was attributed to Intra-Class Correlation variation (i.e., ICC = 0.40). The variation in the ICC decreased to 13% (0.13) in Model 1 (individual-level variables only), whilst in the household-level variables only (Model 2), the ICC increased to 12% (ICC = 0.12). In Model 3, with only community-level variables, the ICC was 0.17 showing a difference of 0.05 in comparison to Model 2 (ICC = 0.12). However, adjusting for individual, household, and community factors, Model 4 only explained a 7% variation in CS. This further reiterates that the likelihood of CS is attributed to the clustering variation in PSUs. The Akaike's Information Criterion [31] and Schwarz's Bayesian Information Criteria (BIC) values showed a successive reduction, which means a substantial improvement in each of the models over the previous model and affirmed the goodness of Model 4 developed in the analysis. Therefore, Model 4, the complete model with the selected individual and household/community factors, was selected to predict the likelihood of CS.

## Discussion

This study examined the prevalence and multilevel factors associated with CS among women of reproductive age in Nigeria. The results revealed low use of CS in Nigeria and that several individual and community-level factors such as age, religious belief, number of children, birth weight, ethnicity, place and region of residence were associated with CS utilisation in Nigeria.

This study found a CS use prevalence of 3.11%, which is slightly higher than the CS prevalence reported in previous studies [4, 6]. The differences in our findings may be as a result of methodological differences. Still, the prevalence (3.11%) found in this study is well below the WHO optimal recommendation of 10–15% [2]. Overall, our study demonstrated an urgent need to implement strategies targeted at increasing the equitable utilisation of maternal health-care services and address barriers that affect its utilisation.

Age was a significant predictor of CS in this study, as seen in models 1 and 4. This result is consistent with previous studies [4, 6, 20] and suggests that older-aged women may be more predisposed to maternal complications and other comorbidities, which may require CS than their younger counterparts [32]. Given that younger-aged pregnancy and birth are also associated with increased obstetric care needs [33], this study calls for awareness of and access to proper ANC, quality healthcare facilities, and skilled birth attendants.

Another important finding in this study is the almost four-fold increase in CS utilisation among women who attended four or more ANC visits than those who did not. This corroborates with previous study [4] and demonstrates the importance of ANC in providing timely obstetric interventions for high-risk pregnancies [23]. Beyond the benefits of ANC on CS utilisation, ANC remains one of the most promising channels to empower women with accurate information to make an informed decision about their care. As a result, this study calls for continued efforts to prioritise ANC utilisation in closing disparities in CS uptake in Nigeria.

Furthermore, the odds of CS in Nigeria are also high among people living in urban areas with at least secondary education and wealthier. This could be as a result of increased level of exposure and access to information than their rural, non-literate and poor counterparts. This result supports the findings in a study across low- and middle-income countries showing place of residence (i.e., urban residence) and level of education (i.e., secondary education and beyond) were likely predictors of C-section [20, 32]. Findings in other parts of the world have also shown that higher education levels and women within a richer wealth index are positively associated with CS [34]. Our study also suggests that the rural-urban disparity in access to CS may translate to higher maternal mortality rates reported in rural Nigeria than in urban areas [35]. The urban-rural CS prevalence disparity could also be explained by the low care quality, high cost of care, low partner support, and poor physical access to healthcare services in rural communities [36]. Nevertheless, strengthening healthcare facilities and improving access to quality healthcare in rural areas may encourage positive attitudes toward CS, thereby reducing maternal morbidity and mortality in the country.

The study also found that parity (i.e., the number of children ever born) and childbirth weight are important factors that influence CS among women of reproductive age in Nigeria. It is likely that multiparous women are more experienced from previous childbirth, may have previous CS, or may be at higher risk of developing complications requiring more medical interventions and may seek timely care when needed than nulliparous women. In the same vein, normally weighted children are less likely to be delivered through CS than high-weighted children. These findings are consistent with findings from a previous study [20].

Culture and religion play crucial roles in shaping the decisions of individuals and groups. This is vivid as the results showed that women who practised Islam were less likely to adopt a CS, unlike their Christian counterparts. This is consistent with the findings from the previous study and could be due to several longstanding religious beliefs inherent in the Islam religion [4]. This current data also found large disparities in CS utilisation among Nigeria's three major and other minority ethnic groups. The northern region, predominantly Hausa/Fulani, was less likely to utilise a CS than the Southern region (Yoruba and Igbo ethnic groups). Though early marriage in the northern region may expose young girls and women to complications, which might require CS, cultural practices also prevent pregnant women from seeking care or showing distress from pregnancy [37, 38]. This is worsened by the fact that most women of the Hausa/Fulani ethnic origin do not have formal education compared to other groups [39, 40].

Mass media has also been essential in improving and disseminating public health knowledge and information and changing health behaviours, especially in rural areas [41, 42]. Thus, our findings showed a strong association between CS and people who were more exposed to mass media and also had high literacy levels. Our findings indicate that increasing mass media and literacy levels help to dispel false cultural beliefs and misconceptions. They are likely to be effective channels for improving CS uptake and providing awareness towards influencing the decisions of pregnant women [43].

Overall, this study aligns with Andersen-Newman's healthcare utilisation theory positing that individual (e.g., maternal age, ethnicity, parity, religion, child birth weight, and antenatal care visits) and societal/communal factors (e.g., household/community-level factors such as

place of residence, wealth index, region, and community literacy level) influence healthcare access and uptake [17, 44]. Based on Newman's theoretical framework, our results indicate a confluence of determinants associated with CS uptake in Nigeria; as such, interventions and programs should adopt a holistic multipronged strategy to increase CS uptake in Nigeria to ultimately improve maternal and newborn health and health outcomes in Nigeria.

## Strengths and limitations

This study utilises a nationally representative sample to examine the prevalence, individual, and household/ community level determinants associated with CS in Nigeria using multi-level analysis. However, there are noteworthy limitations. A major limitation is that the estimate of other pertinent variables, such as pregnancy complications driving the demand for CS, were not taken into consideration. Additionally, we were unable to ascertain whether the CS reported was elective or emergency. Also, as with every other cross-sectional study, this analysis is prone to recall and misclassification bias. We could not also accurately establish causality between the outcome and independent variables due to cross-sectional nature of the survey. The temporality of cross-sectional studies, which allows data to be collected within a short period, could make researchers omit to measure certain variables. Lastly, variables such as place of delivery and birth attendant classification were not included in the study because the DHS "KR file" utilised in this study showed high missing values. Despite these limitations, the NDHS data is a highly cited scientific survey that provides evidence-based and generalisable estimates across a broad range of health indicators. Moreover, our findings analysed the individual and community level factors associated with CS, which provides additional insight into the factors affecting its use.

## Conclusions and recommendations

Given the findings of this study, it is clear that there is a need for the inclusion of different regional, local, and cultural contexts in developing evidence-based policy and programmatic efforts. Our study reveals the need to: improve the quality of education disseminated through the mass media and during antenatal care visits, discourage child marriage, and promote universal basic education and universal health care access through the National Health Insurance Scheme (NHIS), as this may help address the gaps in service coverage. Lastly, this study provides relevant information to inform obstetric decision-making on CS.

## Acknowledgments

The authors are grateful to MEASURE DHS for granting access to the dataset used in this study.

## Author Contributions

**Conceptualization:** Kobi V. Ajayi.

**Data curation:** Samson Olowolaju, Obasanjo Afolabi Bolarinwa.

**Formal analysis:** Samson Olowolaju, Obasanjo Afolabi Bolarinwa.

**Investigation:** Kobi V. Ajayi, Obasanjo Afolabi Bolarinwa.

**Methodology:** Kobi V. Ajayi, Obasanjo Afolabi Bolarinwa.

**Project administration:** Kobi V. Ajayi.

**Resources:** Obasanjo Afolabi Bolarinwa.

**Supervision:** Obasanjo Afolabi Bolarinwa.

**Validation:** Obasanjo Afolabi Bolarinwa.

**Visualization:** Obasanjo Afolabi Bolarinwa.

**Writing – original draft:** Kobi V. Ajayi, Samson Olowolaju, Yusuf H. Wada, Sonya Panjwani, Bright Ahinkorah, Abdul-Aziz Seidu, Collins Adu, Olajumoke Tunji-Adepoju, Obasanjo Afolabi Bolarinwa.

**Writing – review & editing:** Kobi V. Ajayi, Samson Olowolaju, Yusuf H. Wada, Sonya Panjwani, Bright Ahinkorah, Abdul-Aziz Seidu, Collins Adu, Olajumoke Tunji-Adepoju, Obasanjo Afolabi Bolarinwa.

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
