## [Decision Letter · Decision Letter 0]

10 Aug 2022

PGPH-D-22-00815

Prevalence and Factors Associated with Caesarean Section Delivery among women of reproductive age in Nigeria: A Multi-level Analysis of the 2018 Nigeria Demographic and Health Survey

Dear Dr. Bolarinwa,

Thank you for submitting your manuscript to PLOS Global Public Health. After careful consideration, we feel that it has merit but does not fully meet PLOS Global Public Health’s publication criteria as it currently stands. Therefore, we invite you to submit a revised version of the manuscript that addresses the points raised during the review process.

Please note that we have only been able to secure a single reviewer to assess your manuscript. We are issuing a decision on your manuscript at this point to prevent further delays in the evaluation of your manuscript. Please be aware that the editor who handles your revised manuscript might find it necessary to invite additional reviewers to assess this work once the revised manuscript is submitted. However, we will aim to proceed on the basis of this single review if possible. 

We look forward to receiving your revised manuscript.

Kind regards,

Julia Robinson

Staff Editor

Journal Requirements:

Additional Editor Comments (if provided):

Reviewers' comments:

Reviewer's Responses to Questions

**Comments to the Author**

1. Does this manuscript meet PLOS Global Public Health’s publication criteria? Is the manuscript technically sound, and do the data support the conclusions? The manuscript must describe methodologically and ethically rigorous research with conclusions that are appropriately drawn based on the data presented.

Reviewer #1: No

2. Has the statistical analysis been performed appropriately and rigorously?

Reviewer #1: No

3. Have the authors made all data underlying the findings in their manuscript fully available (please refer to the Data Availability Statement at the start of the manuscript PDF file)?

Reviewer #1: Yes

4. Is the manuscript presented in an intelligible fashion and written in standard English?

Reviewer #1: Yes

5. Review Comments to the Author

Reviewer #1: The manuscript examined caesarean delivery in Nigeria using individual and community characteristics as the explanatory factors. This is an important public health concern in the country due to the steady rise in the demand for CS despite the poor state of obstetrics and emergency care in the country. The study was well conceptualized but poorly executed. The selection and recoding of the variables need modification. The findings were adequately reported. There are no serious ethical issues raised by the study. I have no evidence that the study is plagiarized work. The writing style of the authors is satisfactory but the manuscript in its current version needs substantial revision. Specific comments on the sections are outlined below:

Abstract

The abstract is informative but the background section needs to show the knowledge gap study intends to fill.

Introduction

a. Authors wrongly claimed that no studies have examined the multilevel factors predicting CS utilization in Nigeria. For instance, see https://doi.org/10.1080/07399332.2018.1443107 and https://doi.org/10.5281/zenodo.6446080. Authors should examine these studies to properly identify a knowledge gap to be filled by their study.

b. The authors should identify existing policy/strategies that their findings may inform

c. The study should be guided by an appropriate research question or hypothesis

d. If possible, authors should identify a theory or analytical model that could underpin the study

Methods

a) Authors should describe the exclusion criteria

b) Outcome variable: the authors should describe how the variable was coded to help readers know the category of interest. If possible, authors should cite studies that adopted similar operationalization

c) Individual-level explanatory variable: (i) it does not look good to group secondary and higher education together. Evidence shows that secondary education is the dominant level attained by women in the country. The few that attained higher education may show different socioeconomic status and characteristics (ii) authors should have focused on the major ethnic group of Hausa/Fulani, Igbo, and Yoruba while excluding minority groups. It is offensive to describe separate nationalities as ‘Others’. In addition, this group of ‘others’ has different socio-cultural practices that may affect their perception and utilization of CS (iii) child sex and size should have been control variables because they are not individual characteristics of the women used as a unit of analysis

d) Community-level explanatory variables: (i) household head sex is not a community characteristic. Attention should be paid to the fact that individuals are nested in households while households are nested in the community

e) Data analysis: this section should describe how the importance of the community-level characteristics was estimated

f) Describe how variables were selected for the multilevel model. Also, describe the essence of each model (what do they control for?)

Results

a. As shown in Table 2, why are household wealth and sex of head of household presented as contextual factors instead of household factors? The two variables were not aggregated

b. The reported level of secondary/higher education (67.84%) is bogus because of the way the variable was recoded. I doubt if any study has ever found such in the country.

c. Results of the fixed effects should be presented model-by-model as done for the random effects so that readers can follow the importance of variables being added into successive models. Alternatively, the authors may state in the preceding section that the full model is the main model that will be explained

Discussion

a. Authors suggested the “need to multipronged implement strategies to increase CS delivery and to address barriers that affect its utilization” While I agree that the barriers should be addressed among those who demanded it, I wish to know if increasing CS delivery should be a goal in the absence of a medical recommendation

6. PLOS authors have the option to publish the peer review history of their article (what does this mean?). If published, this will include your full peer review and any attached files.

**Do you want your identity to be public for this peer review?** For information about this choice, including consent withdrawal, please see our Privacy Policy.

Reviewer #1: **Yes: **Bola Lukman Solanke

---

## [Decision Letter · Decision Letter 1]

7 Oct 2022

PGPH-D-22-00815R1

Prevalence and Factors Associated with Caesarean Section Delivery among women of reproductive age in Nigeria: A Multi-level Analysis of the 2018 Nigeria Demographic and Health Survey

Dear Dr. Bolarinwa,

Thank you for submitting your manuscript to PLOS Global Public Health. After careful consideration, we feel that it has merit but does not fully meet PLOS Global Public Health’s publication criteria as it currently stands. Therefore, we invite you to submit a revised version of the manuscript that addresses the points raised during the review process. 

We look forward to receiving your revised manuscript.

Kind regards,

Stephen J. McCall, DPhil

Academic Editor

Journal Requirements:

Additional Editor Comments (if provided):

Reviewers' comments:

Reviewer's Responses to Questions

**Comments to the Author**

1. If the authors have adequately addressed your comments raised in a previous round of review and you feel that this manuscript is now acceptable for publication, you may indicate that here to bypass the “Comments to the Author” section, enter your conflict of interest statement in the “Confidential to Editor” section, and submit your "Accept" recommendation.

Reviewer #1: All comments have been addressed

Reviewer #2: (No Response)

Reviewer #3: (No Response)

2. Does this manuscript meet PLOS Global Public Health’s publication criteria? Is the manuscript technically sound, and do the data support the conclusions? The manuscript must describe methodologically and ethically rigorous research with conclusions that are appropriately drawn based on the data presented.

Reviewer #1: Yes

Reviewer #2: Partly

Reviewer #3: Yes

3. Has the statistical analysis been performed appropriately and rigorously?

Reviewer #1: Yes

Reviewer #2: No

Reviewer #3: Yes

4. Have the authors made all data underlying the findings in their manuscript fully available (please refer to the Data Availability Statement at the start of the manuscript PDF file)?

Reviewer #1: Yes

Reviewer #2: (No Response)

Reviewer #3: Yes

5. Is the manuscript presented in an intelligible fashion and written in standard English?

Reviewer #1: Yes

Reviewer #2: Yes

Reviewer #3: Yes

6. Review Comments to the Author

Reviewer #1: 1. The statement on Lines 310-313 is not completed

2. Check the theory mentioned in Line 310. It ought to be Andersen-Newman

Reviewer #2: 1. SUMMARY OF THE ARTICLE:

The study assessed the national prevalence of CS uptake in Nigeria using DHS 2018 cross-sectional data, and estimated the main effect of associated factors at the individual and supra-individual levels, adjusted for covariates. Being of Yoruba ethnicity, parity>=2, practicing Islam, “average” birth weight baby, rural residency, and being from the south-south region of Nigeria predicted less likelihood of CS uptake when compared with Hausa women, those with parity=1, Christians, having macrosomic babies, living in urban areas and the North Central.

The authors chose a valid research question of contemporary public health importance to the research setting, and fits within the context of existing literature in related discipline. Hence, the findings of the study will add value to the body of knowledge, which could influence further studies and policy action. The study design, sample size and choice of multi-level analysis are apt but, the conceptual definition, delineation and classification of what constitute the multi-levels and factors in each level was seemingly defective and poorly executed. I will recommend that a major revision of the article with the multi-level analysis repeated before acceptance for publication.

2. MAJOR ISSUES:

The study was designed to do a multilevel analysis. As defined by the authors, the individual, household, and community levels are not too many to justify any need to collapse household and community factors together. Theoretically, household factors and community factors are distinct enough to warrant delineation. It will be useful for the authors to show effects of factors at the household level differently from those at the community level, before the final model of all factors.

More importantly, the authors need to meticulously ensure that factors selected in each level are mutually exclusive to avoid multicollinearity. With multicollinearity, the multi-level effect size estimates (regression coefficients) will be biased as estimates of the standard errors and variance inflation factors (VIF) are affected. This is a major problem with the current analysis done by the authors e.g., they selected education, occupation and employment status at the individual level while, wealth index, community literacy level, and community-level of socioeconomic status was selected at the second level. Correlation between these factors is apparent, which implies inherent multicollinearity. Authors should revise the classification of variables at each level and repeat the multi-level model building

Other major issues are highlighted below under the various sections in which they occurred:

Background:

- Authors needs to highlight in one-two statements, some theoretical framework in literature for the multi-level approach chosen or state how uptake of CS or other related maternal health services have been shown to be determined differently by factors at the various levels. This study is not the first to think this way. The Newman’s healthcare utilization theory mentioned in the discussion should have been mentioned here at first as the background to their chosen approach.

Methods (variable Definition and Data Analysis):

- The authors should consider replacing the term “child size” with the more specific term “child birth weight” because size could imply other anthropometric measures like length/height. Also, birth weight is usually classified as low, normal or high. There is a standard definition of those. The authors used small, average and large without defining what this means, if any different from low, normal & high birth weight

- The authors merged Hausa and Fulani into one category. The name of the final category should reflect that it includes Fulani women, and not just Hausa. For the regression analysis, it may be more logical to “others” the reference category, so you can compare the hausa, fulani and igbo to others.

3. MINOR ISSUES:

It is a well written article overall but, there are many grammatical and typographical errors throughout the article, which can be easily corrected if the [many] authors will consider proof-reading carefully or have a language expert do that for them.

The Title: it is unnecessary verbose with no added information of some included/repeated words. E.g., the phrase “among women of reproductive age” can be excluded since CS delivery is typically expected in pregnant women that are often of reproductive age. Moreso that DHS is on the title, and DHS usually collects data from that age-group. Also, Nigeria did not need to appear twice in that title. Consider revising the title.

The Abstract: Authors should consider revising the entire background statement for grammar, tenses and flow of thought. “Vaginal delivery” mentioned at the beginning has questionable value as the article is entirely about CS delivery. Also, “maternal health” does not include “neonatal health”. Authors should revise that statement. The methods and result section under abstract should be re-written after authors have revised the article and repeated the multi-level analysis. The conclusion should be revised to reflect the specific findings of this study and not what the authors wish, generally.

4. OTHER POINTS (optional notes):

- The tables could be better formatted and improved. Authors can look up similar published tables for guides

- The discussion, conclusion and recommendations could be more robust, systematic, and specific to the findings of this study.

Reviewer #3: I have provided comments to the authors in the attached review. Thank you.

7. PLOS authors have the option to publish the peer review history of their article (what does this mean?). If published, this will include your full peer review and any attached files.

**Do you want your identity to be public for this peer review?** For information about this choice, including consent withdrawal, please see our Privacy Policy.

Reviewer #1: **Yes: **Bola Lukman Solanke

Reviewer #2: **Yes: **Abraham Braimah Idokoko

Reviewer #3: No

---

## [Editor Report · Decision Letter 2]

24 Nov 2022

PGPH-D-22-00815R2

A multi-level analysis of prevalence and factors associated with Caesarean Section delivery in Nigeria

Dear Dr. Bolarinwa,

Thank you for submitting your manuscript to PLOS Global Public Health. After careful consideration, we feel that it has merit but does not fully meet PLOS Global Public Health’s publication criteria as it currently stands. Therefore, we invite you to submit a revised version of the manuscript that addresses the points raised during the review process.

We look forward to receiving your revised manuscript.

Kind regards,

Stephen J. McCall, DPhil

Academic Editor

Journal Requirements:

Additional Editor Comments (if provided):

Major issues:

The model building strategy could be slightly clearer and systematic. How were variables added to the model and how did you chose if they were included in the final model?

There is a good summary on model selection techniques: Laubach, Zachary M., et al. "A biologist's guide to model selection and causal inference." Proceedings of the Royal Society B 288.1943 (2021): 20202815. [Pre-print version (open access): https://arxiv.org/ftp/arxiv/papers/2010/2010.07506.pdf]

Ultimately - what is the aim of the model building strategy is it to obtain a predictive model (identify a set of variables that best explain the variation in the outcome) or is it to understand the relationship between each factor and the outcome? If it the latter each exposure should be modelled separately with the outcome due to the table 2 fallacy:

Westreich, Daniel, and Sander Greenland. "The table 2 fallacy: presenting and interpreting confounder and modifier coefficients." American journal of epidemiology 177.4 (2013): 292-298.

If it is to build a predictive model then the PROBAST, TRIPOD guidelines require to be followed. Also the interpretation of the adjusted coefficients requires to be removed as they are not interpretable due to the table 2 fallacy.

Minor changes required:

Abstract: remove the word predict and the statistical test from the conclusion. Rather try and conclude about what the results show?

Throughout I suggest to use the term "use of casearean section" rather than cesearean section rates - as technically cs rate is not a rate although it is prevalent in the literature.

Since you are publishing in PLOS there is no word count so remove the abbreviation CS through and replace with unabbreviated form.

Line 126 "Specifically, for this study, we used the “KR file” of the 2018 NDHS as recommended by analysts of the Demographic Health Survey" No need to mention the KR - we will assume you have used the correct file.

Line 137-143 - is this C-S at last birth?

Line 157-161: For the score you created what was the cohen's kappa?

Line 201: What number did you consider as collinear?

First paragraph of the discussion should present the main findings of the study only. Then the proceeding paragraph should place the findings in context.
---

## [Decision Letter · Decision Letter 3]

12 Apr 2023

PGPH-D-22-00815R3

A multi-level analysis of prevalence and factors associated with Caesarean Section delivery in Nigeria

Dear Dr. Bolarinwa,

Thank you for submitting your manuscript to PLOS Global Public Health. After careful consideration, we feel that it has merit but does not fully meet PLOS Global Public Health’s publication criteria as it currently stands. Therefore, we invite you to submit a revised version of the manuscript that addresses the points raised during the review process.

Your manuscript has been reassessed by two reviewers from an earlier round, whose reports can be found below. As you will see from the comments, the reviewers acknowledge that the manuscript has improved significantly, but there remain some minor concerns which should be addressed before your manuscript is suitable for publication.

We look forward to receiving your revised manuscript.

Kind regards,

Dr Joseph Donlan

Senior Editor

Journal Requirements:

Additional Editor Comments (if provided):

Reviewers' comments:

Reviewer's Responses to Questions

**Comments to the Author**

1. If the authors have adequately addressed your comments raised in a previous round of review and you feel that this manuscript is now acceptable for publication, you may indicate that here to bypass the “Comments to the Author” section, enter your conflict of interest statement in the “Confidential to Editor” section, and submit your "Accept" recommendation.

Reviewer #2: All comments have been addressed

Reviewer #3: All comments have been addressed

2. Does this manuscript meet PLOS Global Public Health’s publication criteria? Is the manuscript technically sound, and do the data support the conclusions? The manuscript must describe methodologically and ethically rigorous research with conclusions that are appropriately drawn based on the data presented.

Reviewer #2: Yes

Reviewer #3: Yes

3. Has the statistical analysis been performed appropriately and rigorously?

Reviewer #2: Yes

Reviewer #3: Yes

4. Have the authors made all data underlying the findings in their manuscript fully available (please refer to the Data Availability Statement at the start of the manuscript PDF file)?

Reviewer #2: Yes

Reviewer #3: Yes

5. Is the manuscript presented in an intelligible fashion and written in standard English?

Reviewer #2: Yes

Reviewer #3: Yes

6. Review Comments to the Author

Reviewer #2: The manuscript assessed the prevalence of CS uptake in Nigeria using DHS 2018 data and attempted to determine the multilevel factors associated with CS uptake. The authors chose a valid research question of contemporary public health importance to the research setting and fits within the context of existing literature in related discipline. Hence, the findings of the study will add value to the body of knowledge, which could influence further studies and policy action. The study design, sample size and choice of multi-level analysis are apt.

The authors made adequate attempt at resolving most of the concerns raised in revision 2, except for their choosing not to disaggregate household and community factors. Considering the current state of the article, I will recommend its acceptance for publication, provided that the authors are made to correct the typographical errors that persist in the paper. Also, they should consider reformatting the tables.

Reviewer #3: Review of manuscript entitled: “A multi-level analysis of prevalence and factors associated with Caesarean Section delivery in Nigeria”

Thank you for the opportunity to review this manuscript again. This study uses a multilevel analysis to assess the relationship between variables at the individual level and household/community level and cesarean section deliveries in Nigeria using a nationally representative household survey. The findings of this study provide important insights into the characteristics of women and the households/communities they reside in and how these characteristics differ for women who delivered their last birth as a cesarean section versus normal vaginal delivery.

The authors have attended to the reviewer’s comments and this version reads very well. The manuscript clearly presents the results and discusses the findings in a comprehensive way. The limitations of the study are well articulated.

Some comments, mainly minor, are listed below:

• On p.16, the results section “Random effects” starts by referring to the variation based on the clustering level of the MLM. And then you change to referring to the ICC. You interchange between the terms variation and ICC and I think it should be written in a clearer way.

• With regards to the ICC, it would be useful to give a brief description in the methodology about how you estimated this value since in logistic MLMs it is not directly possible to generate an ICC.

• On p.9, line 186: there is a typo with regards to the value of the p-value.

• On p.9, line 191: the sentence wording needs to be revisited.

• On p.26, line 299 there is another typo in the sentence.

Thank you!

7. PLOS authors have the option to publish the peer review history of their article (what does this mean?). If published, this will include your full peer review and any attached files.

**Do you want your identity to be public for this peer review?** For information about this choice, including consent withdrawal, please see our Privacy Policy.

Reviewer #2: **Yes: **Abraham Braimah Idokoko

Reviewer #3: No

---

## [Editor Report · Decision Letter 4]

4 May 2023

A multi-level analysis of prevalence and factors associated with Caesarean Section in Nigeria

PGPH-D-22-00815R4

Dear Dr. Bolarinwa,

We are pleased to inform you that your manuscript 'A multi-level analysis of prevalence and factors associated with Caesarean Section in Nigeria' has been provisionally accepted for publication in PLOS Global Public Health.

Best regards,

Julia Robinson

Executive Editor